# Risk association of *RANKL* and *OPG* gene polymorphism with breast cancer to bone metastasis in Pashtun population of Khyber Pakhtunkhwa, Pakistan

**Faiza Hayat[1], Najeeb Ullah Khan[1]*, Aakif Ullah Khan[2], Iftikhar Ahmad[2], Ahmad M. Alamri[3]*, Bushra Iftikhar[4]***

**1** Institute of Biotechnology and Genetic Engineering (Health Division), The University of Agriculture, Peshawar, Khyber Pakhtunkhwa, Pakistan, **2** Institute of Radiotherapy and Nuclear Medicine (IRNUM), Peshawar, Khyber Pakhtunkhwa, Pakistan, **3** Research Center for Advanced Materials Science (RCAMS), King Khalid University, Abha, Saudi Arabia, **4** Khyber Medical College Peshawar, Peshawar, Pakistan

* najeebkhan@aup.edu.pk (NUK); aalamri@kku.edu.sa (AMA); bushraiftikhar@hotmail.com (BI)

## Abstract

### Introduction

The receptor activator NF-κB ligand (*RANKL*) and Osteoprotegrin (*OPG*) single nucleotide polymorphisms (SNPs) have been associated with the risk of breast cancer to bone metastasis. This study was designed to investigate the association of *RANKL* and *OPG* gene polymorphisms with breast to bone metastasis in Pashtun population of Khyber Pakhtunkhwa, Pakistan.

### Materials and methods

A total of 215 participants were enrolled containing 106 breast cancer patients, 58 breast to bone metastasis and 51 age and gender matched healthy controls. *RANKL* (rs9533156) and *OPG* (rs2073618, rs3102735) polymorphisms were genotyped in genomic DNA, using Tetra-ARMS PCR protocol. The results were analyzed among the three groups and *P*-value less then 0.05 were considered statistically significant.

### Results

Our results displayed significant association of *OPG* (rs3102735) risk allele and corresponding genotypes in breast cancer vs healthy controls, bone metastasis vs healthy controls and breast cancer vs breast to bone metastasis as a disease risk. However, there was no association observed for *OPG* (rs2073618) risk allele and corresponding genotypes with the diseases risk. Similarly, *RANKL* (rs9533156) risk allele and corresponding genotypes in breast cancer vs healthy controls, bone metastasis vs healthy controls and breast cancer vs breast to bone metastasis exhibited significant association except for the risk allele carrying genotypes in breast to bone metastasis.

**Data Availability Statement:** All relevant data are within the paper and its Supporting information files.

**Funding:** The authors appreciate the support of the Research Center for Advanced Materials Science (RCAMS) at King Khalid University Abha, Saudi Arabia through a project number RCAMS/KKU/22. The funders had no role in study design, data collection and analysis, decision to publish, or preparation of the manuscript.

**Competing interests:** The authors have declared that no competing interests exist.

## Conclusion

*OPG* (rs3102735) and *RANKL* (rs9533156) exhibited significant association with breast to bone metastasis while *OPG* (rs2073618) didn't show significant association with breast to bone metastasis in Pashtun population of Pakistan. However, this study unlocks more questions to investigate the exact scenario of genetic predisposition of breast to bone metastasis.

## Introduction

Breast cancer is the most frequently diagnosed cancer in women worldwide [1] and one of the leading cancers in Asian countries, including Pakistan [2]. Breast cancer is regularly metastasizing to various organs such as bone (60–80%), lung (57–77%), liver (5–12%) and brain (10–30%) [3–6]. The process of metastasis is a multi-step sequential cascade starting primary tumor local invasion, progression towards lymphogenic spread and colonization at distant sites [7]. Genetic factors may predispose the probability of breast cancer metastasize to bone [8]. The discovery of the apoptosis regulator receptor activator NF-κB ligand (*RANKL*) and osteoclastogenesis inhibitor *OPG* signaling pathway exhibited major regulatory system for osteoclast formation and also *OPG/RANKL* alliance controlled the bone remodeling [9, 10], contributed significantly to the physiological bone turnover and also involved in osteoclast differentiation and activation [11].

*RANKL-OPG* pathway contributes to the primary tumorigenesis, bone metastasis and regulated by factors including progesterone and prolactin. Skeletal diseases caused by excessive osteoclastic activity including breast cancer to bone metastasis, rheumatoid arthritis and osteoporosis were triggered by imbalance between *RANKL* and *OPG* genes [12–15]. In addition, the *RANKL* and *OPG* also involved in proliferation of malignant and benign breast tumors [9]. Over expression of *OPG* by cancerous cells, increases tumor growth and cell proliferation [16]. *OPG* genetic variants are associated with bone mineral density, bone turnover and osteonecrosis [17–19].

*RANKL* polymorphism located in the proximal promoter interacts with transcription factors such as heat shock proteins, E2F1 (E2F transcription factor 1), vitamin D3 and core-binding factor a1 (Cbfa1), which affect the expression of *RANKL* gene [20, 21]. *RANKL* rs9533156 (T>C) causes DNA hypermethylation leading to gene silencing and transcriptional inactivation, that affects the binding of transcriptional factors [22, 23]. *OPG* rs2073618 influence secretions from the cells and drastically affect the secretory kinetics [10].

*RANKL/OPG* pathway suggests an important effect on the pathogenesis of breast cancer due to functional properties of genes. Genetic variation within *OPG* (rs3102735) promoter region and *RANKL* (rs9533156) near 5' region could have an effect on the expression of genes and may have an influence on the development of tumors [10, 12]. Various studies conducted have elucidated the association of *RANKL* and *OPG* polymorphism with the breast cancer to bone metastasis risk in different ethnicities [24]. However, unfortunately, there is no such work done on Pakistani population practically the Pashtun population of Khyber Pakhtunkhwa, Pakistan. Here this study aimed to investigate the association of above-mentioned polymorphisms with breast cancer to bone metastasis in the patients of Pashtun population of Khyber Pakhtunkhwa, Pakistan.

## Materials and methods

### Study design, ethical approval and sampling

This case-control study was conducted at the Institute of Biotechnology and Genetic Engineering (IBGE), The University of Agriculture Peshawar (UAP), Pakistan. Patients were recruited from the Institute of Radiotherapy and Nuclear Medicine (IRNUM) Peshawar, Pakistan. The study protocol and informed consent (patients proforma) were approved (IBGE/UAP/2020/003) by the Ethical Committee of IBGE, UAP. Written informed consent was taken from all the participants while explaining the aim and objectives of the study. A total of 215 participants were enrolled, with age ranging from 25–80 years, containing (106 breast cancer patients only, 58 breast cancer to bone metastasis, and 51 healthy controls) from January 2021–December 2021. The demographic, clinical parameters and blood samples were collected from all the subjects through venipuncture procedure using sterile syringe. The blood was stored at– 20ºC in EDTA tubes at IBGE until further process.

### DNA isolation, SNP genotyping and gel electrophoresis

DNA was extracted from the whole blood using salting out (non-enzymatic) method and SNP genotyping was confirmed via T-ARMS PCR, previously adopted in our lab [25, 26]. Specific primers were designed using primer blast software for each SNP (Table 1). The PCR mixture (10μL) was prepared, consisting of 5μL master mix (Thermo Fisher Scientific, DreamTaq Master Mix (2X), 0.5μL of each forward and reverse primer, 3μL of ddH$_2$O and 1μL of template DNA. The PCR condition were, initial denaturation at 95˚C for 5min. followed by 35 cycles, denaturation at 95˚C for 30 sec. annealing for 30 sec. at 61˚C, 63˚C and 60˚C for *OPG* rs2073618, rs3102735 and *RANKL* rs9533156 respectively, extension 72˚C for 30 sec. and final extension at 72˚C for 7min. The amplified PCR products were run and confirmed using 1.5% gel electrophoresis with 1KB DNA ladder (Thermo Fisher Scientific).

### Statistical analysis

SPSS software (version 2.0) was used for statistical analysis. Demographic data was analyzed via descriptive statistical method and $p < 0.05$ was considered significant. Chi-square test was performed for categorical data, while t-test and one-way ANOVA was applied for analysis of quantitative data. The genotypic and allelic frequency was calculated by using chi-square and

**Table 1. Specific primer sequences used for genotyping.**

| Genes (SNP) | Primers | Sequence 5'-3' |
|---|---|---|
| *OPG* (rs2073618), C/G | Forward outer | AGCCTAACCCCAAGCCTC |
| | Reverse outer | CCCTGGGGGATCCTTTCC |
| | Forward inner | AGCGCGCAGCACAGCAAC |
| | Reverse inner | TCCGGGGACCACAA |
| *OPG* (rs3102735), T/C | Forward outer | CGTACCCGGCTGCCTGAC |
| | Reverse outer | CCGGGTACGGCGGAAACT |
| | Forward inner | GTTCGCTGTCTCCCCCATCTGAACAAC |
| | Reverse inner | GTCTAACTTCTAGACCAGGGAATTA |
| *RANKL* (rs9533156), TC | Forward outer | TCAGCAACTTCCTTCTGAAGAAGT |
| | Reverse outer | AGTCTCGGTTTCCTTAGGATTTAGA |
| | Forward inner | CCCTTTACCCTTTTCTCTGCACC |
| | Reverse inner | CTGACTTTATAAAGATGAAAACTCCA |

Fisher exact test. Online software named Medcalc was used for obtaining odd ratios and 95% CI.

## Results

### Demographic and clinicopathological parameters association with diseases risk

A total of 215 participants were recruited in this study, consisting of (breast cancer patients, BC = 106, breast to bone metastasis, BM = 58, and healthy controls, HC = 51). The demographics and clinical characteristics association was checked among the study groups (Table 2). The results showed no significant association among the ages ($p = 0.811$), tumor size ($p = 0.268$), stage ($p = 0.067$), HER2 status ($p = 0.284$), PR status ($p = 1.00$), ER status ($p = 0.940$) and Luminal subtypes ($p = 0.639$). However, significant association was observed in menstrual status ($p = 0.0001$), nulliparous ($p = 0.0001$), family history ($p = 0.0001$), localization ($p = 0.0001$), Ki67 ($p = 0.0001$), histology ($p = 0.0001$), nodal status ($p = 0.0001$) and IHC subtypes ($p = 0.0001$).

### *OPG* (rs3102735) and *RANKL* (rs9533156) exhibited significant association with breast cancer risk while *OPG* (rs2073618) didn't show association compared with healthy controls

All the samples were analyzed for SNPs genotyping using T-ARMS-PCR, run and confirmed on 2% agarose gel with 1KB DNA ladder (Fig 1). The results indicated that *OPG* (rs2073618) risk allele C (OR: 1.2293 CI: 0.7651–1.9751, $p = 0.3934$) and risk allele containing genotypes CC (OR: 1.7262 CI: 0.6016–4.9532, $p = 0.3101$) and GC (OR: 1.5517 CI: 0.6165–3.9056, $p = 0.3508$) were statistically not significantly associated with breast cancer risk. In contrast, the risk allele of *OPG* (rs3102735) C (OR: 0.4882 CI: 0.2959–0.8057, $p = 0.005$) and respective genotypes CC (OR: 0.2365 CI: 0.0696–0.8032, $p = 0.0208$) and TC (OR: 0.4138 CI: 0.1991–0.8599, $p = 0.0181$) were statistically significant. Similarly, in case of *RANKL* (rs9533156), the frequency of risk allele C (OR: 2.5887 CI: 1.5924–4.2082, $p = 0.0001$) and respective genotypes CC (OR: 8.1818 CI: 2.7796–24.0836, $p = 0.0001$) and TC (OR: 3.7500 CI: 1.4028–10.0243, $p = 0.0084$) were significantly associated with disease risk (Table 3).

### *OPG* (rs3102735) and *RANKL* (rs9533156) exhibited risk association with breast cancer to bone metastasis while *OPG* (rs2073618) didn't show association compared with healthy controls

The genotypic and allelic frequencies of *OPG* and *RANKL* in bone metastasis and healthy controls were investigated (Table 4). The results indicated that *OPG* (rs2073618), risk allele C (OR: 1.0990 CI: 0.6448–1.8731, $p = 0.7282$) and risk allele containing genotypes CC (OR: 1.2963 CI: 0.3961–4.2421, $p = 0.6679$) and GC (OR: 1.3410 CI: 0.4805–3.7422, $p = 0.5752$) didn't show statistically significant association. Unlikely, *OPG* (rs3102735) indicated that risk allele C (OR: 1.8857 CI: 1.1000–3.2327, $p = 0.0211$) and genotypes CC (OR: 4.5714 CI: 1.3382–15.6164, $p = 0.0153$) and TC (OR: 2.4286 CI: 0.9068–6.5041, $p = 0.0775$) were significantly associated with bone metastasis. Similarly, the *RANKL* (rs9533156) risk allele C (OR: 5.000 CI: 2.7280–9.1642, $p = 0.0001$) and homozygous mutant genotype CC (OR: 11.1818 CI: 3.3299–37.5486, $p = 0.0001$) were significantly associated with high risk of breast to bone metastasis but the heterozygous genotype TC (OR: 1.4400 CI: 0.4234–4.8972, $p = 0.5593$) was not associated with disease risk.

**Table 2. Demographics, clinical and pathological parameters and their association with the disease risk.**

| Characteristics | BC n (%) | BM n (%) | HC | *p*-value |
|---|---|---|---|---|
| Total | 106 | 58 | 51 | |
| Age Group (years) | 45.811±10.477 | 55.55±11.948 | 41.6274±6.627 | 0.811 |
| Menstrual Status | | | | |
| Premenopausal | 43 (40.56) | 19 (32.76) | 36 (70.59) | 0.0001 |
| Postmenopausal | 63 (59.44) | 39 (67.24) | 15 (29.41) | |
| Nulliparous | | | | |
| No | 97 (91.51) | 49 (84.48) | - | 0.0001 |
| Yes | 9 (8.49) | 9 (15.52) | | |
| Family History | | | | |
| Negative | 92 (86.79) | 46 (79.31) | - | 0.0001 |
| Positive | 14 (13.21) | 12 (20.69) | | |
| Localization | | | | |
| Left | 51 (48.11) | 37 (63.79) | - | 0.0001 |
| Right | 55 (51.89) | 21 (36.21) | | |
| Ki67 | | | | |
| <14% | 37 (34.91) | 2 (3.45) | - | 0.0001 |
| >14% | 69 (65.09) | 56 (96.55) | | |
| Tumor Size | | | | |
| T1 (<2cm) | 14 (13.21) | 3 (5.17) | - | 0.268 |
| T2 (2.1–5 cm) | 59 (55.66) | 36 (62.07) | | |
| T3 (>5 cm) | 33 (31.13) | 19 (32.76) | | |
| Stage | | | | |
| 2 | 2 (1.87) | 0 (0) | - | 0.067 |
| 3 | 33 (31.13) | 12 (20.69) | | |
| 4 | 71 (66.98) | 46 (79.31) | | |
| Histology | | | | |
| ILC | 6 (5.66) | 15 (25.86) | - | 0.0001 |
| IDC | 100 (94.34) | 43 (74.14) | | |
| Her 2 Status | | | | |
| Negative | 64 (60.38) | 30 (51.72) | - | 0.284 |
| Positive | 42 (39.62) | 28 (48.28) | | |
| PR Status | | | | |
| Negative | 53 (50) | 29 (50) | - | 1.00 |
| Positive | 53 (50) | 29 (50) | | |
| ER Status | | | | |
| Negative | 50 (40.17) | 27 (46.55) | - | 0.940 |
| Positive | 56 (52.83) | 31 (53.45) | | |
| Nodal Status | | | | |
| Negative | 38 (35.85) | 12 (20.69) | - | 0.0001 |
| Positive | 68 (64.15) | 46 (79.31) | | |
| IHC Subtypes | | | | |
| Luminal A | 35 (33.02) | 15 (25.86) | - | 0.0001 |
| Luminal B | 23 (21.69) | 15 (25.86) | | |
| Her2 enriched | 20 (18.87) | 16 (27.59) | | |
| TNBC | 28 (26.42) | 12 (20.69) | | |
| Luminal subtypes | | | | |
| Luminal A + B | 58 (54.72) | 30 (51.72) | - | 0.639 |

(*Continued*)

**Table 2.** (Continued)

| Characteristics | BC n (%) | BM n (%) | HC | *p*-value |
|---|---|---|---|---|
| Non-Luminal | | | | |
| Her2 enriched + TNBC | 48 (45.28) | 28 (48.27) | - | |

### *OPG* (rs3102735) and *RANKL* (rs9533156) exhibited association with breast to bone metastasis compared with breast cancer patients

The frequencies of *OPG* and *RANKL* in breast cancer and breast to bone metastasis were also checked (Table 5). The results displayed that *OPG* (rs2073618) risk allele C (OR: 1.1186 CI: 0.7094–1.7639, *p* = 0.6295) and risk allele containing genotypes CC (OR: 1.3316 CI: 0.4647–3.8155, *p* = 0.5938) and GC (OR: 1.1571 CI: 0.4548–2.9440, *p* = 0.7593) were not associated with breast to bone metastasis compared with breast cancer patients. Conversely, *OPG* (rs3102735) risk allele C (OR: 0.1704 CI: 0.0716–0.4053, *p* = 0.0001) and genotypes CC (OR: 0.0517 CI: 0.0157–0.1708, *p* = 0.0001) and TC (OR: 0.1704 CI: 0.0716–0.4053, *p* = 0.001) were were strongly associated with breast to bone metastasis. Likewise, *RANKL* (rs9533156) risk allele C (OR: 0.5177 CI: 0.2994–0.8953 *p* = 0.0185) was found significant with disease risk, while the genotypes CC (OR: 0.7317 CI: 0.2220–2.4114, *p* = 0.6077) and TC (OR: 2.6042 CI: 0.7221–9.3922, *p* = 0.1436) were not associated with diseases risk.

## Discussion

Breast cancer is also one of the frequently diagnosed cancers in woman of Pashtun population of Khyber Pakhtunkhwa, Pakistan. Unfortunately, there is very little work conducted so far, to explore the genetic predisposition of breast cancer and associated metastasis in this region.

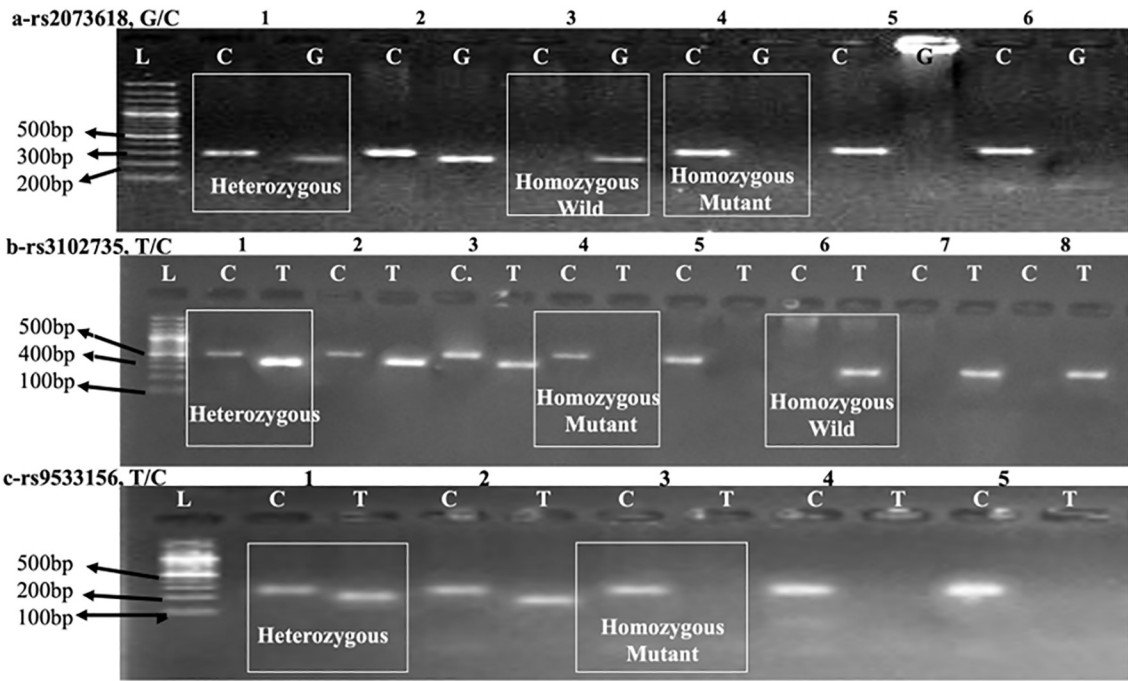

**Fig 1. Electrophoretogram showing T-ARMS PCR based amplification of random samples (a)** *OPG* **(rs2073618), (b)** *OPG* **(rs3102735), (c)** *RANKL* **(rs9533156), L, 1KB DNA ladder (Thermo Fishier Scientific).**

**Table 3. Allelic and genotypic frequencies of *OPG* and *RANKL* in breast cancer vs healthy controls.**

| Genotypes/Alleles | BC | HC | BC vs HC | p-value |
|---|---|---|---|---|
| *OPG* (rs2073618) | n = 106(%) | n = 51(%) | OR 95%CI | |
| GG | 14 (13.21) | 10 (19.60) | | |
| GC | 63 (59.43) | 29 (56.86) | 1.5517(0.6165–3.9056) | 0.3508 |
| CC | 29 (27.35) | 12 (23.53) | 1.7262(0.6016–4.9532) | 0.3101 |
| G | 91 (42.9) | 49 (48.0) | | |
| C | 121 (57) | 53 (51.9) | 1.2293(0.7651–1.9751) | 0.3934 |
| *OPG* (rs3102735) | | | | |
| TT | 58 (54.72) | 16 (31.37) | | |
| TC | 42 (39.62) | 28 (54.90) | 0.4138(0.1991–0.8599) | 0.0181 |
| CC | 6 (5.66) | 7 (13.72) | 0.2365(0.0696–0.8032) | 0.0208 |
| T | 158 (74.5) | 60 (58.8) | | |
| C | 54 (25.0) | 42 (41.0) | 0.4882(0.2959–0.8057) | 0.005 |
| *RANKL* (rs9533156) | | | | |
| TT | 8 (7.55) | 15 (29.41) | | |
| TC | 50 (47.17) | 25 (49.02) | 3.7500(1.4028–10.0243) | 0.0084 |
| CC | 48 (45.28) | 11 (21.57) | 8.1818(2.7796–24.0836) | 0.0001 |
| T | 66 (31.1) | 55 (53.9) | | |
| C | 146 (68.8) | 47 (46.0) | 2.5887(1.5924–4.2082) | 0.0001 |

Here, this study aimed to check the association of *OPG* (rs2073618, rs3102735) and *RANKL* (rs9533156) polymorphism with breast cancer to bone metastasis, as there is no report available for the said population. However, similar studies have been conducted in different ethnicities worldwide to confirm the association of *OPG* and *RANKL* polymorphism with breast cancer to bone metastasis, which displayed contradictory results [24, 27].

**Table 4. Frequencies of *OPG* and *RANKL* polymorphisms in bone metastasis vs healthy controls.**

| Genotypes/Allele | BM | HC | BM vs HC | P-value |
|---|---|---|---|---|
| *OPG* (rs2073618) | n = 58(%) | n = 51(%) | OR (95%CI) | |
| GG | 9 (15.52) | 10 (19.60) | | |
| GC | 35 (60.34) | 29 (56.86) | 1.3410(0.4805–3.7422) | 0.5752 |
| CC | 14 (24.14) | 12 (23.53) | 1.2963(0.3961–4.2421) | 0.6679 |
| G | 53 (45.6) | 49 (48.0) | | |
| C | 63 (54.1) | 53 (51.9) | 1.0990(0.6448–1.8731) | 0.7282 |
| *OPG* (rs3102735) | | | | |
| TT | 8 (13.79) | 16 (31.37) | | |
| TC | 34 (58.62) | 28 (54.90) | 2.4286(0.9068–6.5041) | 0.0775 |
| CC | 16 (27.59) | 7 (13.72) | 4.5714(1.3382–15.6164) | 0.0153 |
| T | 50 (43.1) | 60 (58.8) | | |
| C | 66 (56.8) | 42 (41.0) | 1.8857(1.1000–3.2327) | 0.0211 |
| HWE | 0.331393 | 0.635496 | | |
| *RANKL* (rs9533156) | | | | |
| TT | 5 (8.62) | 15 (29.41) | | |
| TC | 12 (20.69) | 25 (49.02) | 1.4400(0.4234–4.8972) | 0.5593 |
| CC | 41 (70.69) | 11 (21.57) | 11.1818(3.3299–37.5486) | 0.0001 |
| T | 22 (18.9) | 55 (53.9) | | |
| C | 94 (81.0) | 47 (46.0) | 5.000(2.7280–9.1642) | 0.0001 |
| HWE | 0.045103 | 0.995361 | | |

**Table 5. *OPG* and *RANKL* frequencies in breast cancer and breast to bone metastasis.**

| *OPG* (rs2073618) | BC | BM | BC vs BM | *P*-value |
|---|---|---|---|---|
| **Genotypes/Alleles** | **n = 106(%)** | **n = 58(%)** | **OR (95%CI)** | |
| GG | 14 (13.21) | 9 (15.52) | | |
| GC | 63 (59.43) | 35 (60.34) | 1.1571(0.4548–2.9440) | 0.7593 |
| CC | 29 (27.35) | 14 (24.14) | 1.3316(0.4647–3.8155) | 0.5938 |
| G | 91 (42.9) | 53 (45.6) | | |
| C | 121 (57) | 63 (54.1) | 1.1186(0.7094–1.7639) | 0.6295 |
| *OPG* (rs3102735) | | | | |
| TT | 58 (54.72) | 8 (13.79) | | |
| TC | 42 (39.62) | 34 (58.62) | 0.1704(0.0716–0.4053) | 0.001 |
| CC | 6 (5.66) | 16 (27.59) | 0.0517(0.0157–0.1708) | 0.0001 |
| T | 158 (74.5) | 50 (43.1) | | |
| C | 54 (25.0) | 66 (56.8) | 0.1704(0.0716–0.4053) | 0.0001 |
| *RANKL* (rs9533156) | | | | |
| TT | 8 (7.55) | 5 (8.62) | | |
| TC | 50 (47.17) | 12 (20.69) | 2.6042(0.7221–9.3922) | 0.1436 |
| CC | 48 (45.28) | 41 (70.69) | 0.7317(0.2220–2.4114) | 0.6077 |
| T | 66 (31.1) | 22 (18.9) | | |
| C | 146 (68.8) | 94 (81.0) | 0.5177(0.2994–0.8953) | 0.0185 |
| **HWE** | 0.58834 | 0.045103 | | |

Interestingly, our results demonstrated that *OPG* (rs2073618), risk allele C (*p* = 0.3934) and corresponding genotypes GC (*p* = 0.3508 and CC (*p* = 0.3101) didn't establish significant association with breast cancer risk and also no significant correlation was observed between risk allele C (*p* = 0.7282) and risk allele containing genotypes GC (*p* = 0.5752, CC (*p* = 0.6679) with breast to bone metastasis when compared to healthy controls (Tables 3 and 4), which is in contrast to the previous study conducted in Egyptians and Chinese Han population [10, 14, 28, 29]. Another study conducted among Caucasians (rs2073618), which revealed the minor allele to be protective against breast cancer pathogenesis [30]. However, in the present study individuals carrying homozygous mutant genotype (CC) were at increased risk of breast cancer (OR = 1.7262 95% CI = 0.6016–4.9532) and breast to bone metastasis (OR = 1.3316, 95% CI = 0.4647–3.8155) as compared to the individuals carrying heterozygous genotype. Regarding the *OPG* (rs3102735), the results were found significant between risk allele and risk allele containing genotypes with breast cancer and breast to bone metastasis. Our study result is similar to that of the Caucasian population, where the minor allele C and the corresponding genotypes CC and TC were more frequent in breast cancer patients and showed 1.5-fold increased risk [10]. However, more investigation will be required with large data sets to validate the association of *OPG* polymorphism with breast to bone metastasis.

Concerning the *RANKL* (rs9533156), significant association was observed between the risk allele and corresponding genotypes with breast cancer and breast to bone metastasis. Although, in the current study individuals carrying homozygous mutant genotype (CC) were at increased risk of breast cancer (OR = 8.1818 95% CI = 2.7796–24.0836) as compared to breast to bone metastasis (OR = 0.7317, 95% CI = 0.2220–2.4114). The mutant genotype frequency was found higher among breast cancer than bone metastasis patients. Previous study among Egyptians revealed T the major allele frequently among the breast cancer cases, while the present research found minor allele C repeatedly among breast cancer patients when compared to the healthy controls, which is in contrast to the study done in Caucasian and Egyptian

population [10, 14]. A study expressed that minor allele as well as the genotype of the minor allele of the *RANKL* rs9533156 was strongly associated with a higher BMI ($>/ = 28$) patients with breast cancer group. Whether obese patients carrying the minor allele from one of the two *RANKL* SNPs have an additionally a higher risk of developing breast cancer remains open in this study.

Excluding the contradictory results of genetic polymorphism and risk of breast to bone metastasis in various ethnicities, both *OPG* and *RANKL* have important role in breast cancer to bone metastasis proliferation and progression [31]. However, more investigation will be required to see the exact mechanism of both *OPG* and *RANKL* polymorphism and their association with breast cancer to bone metastasis in various population.

## Conclusion

Our study revealed that *OPG* (rs3102735) and *RANKL* (rs9533156) exhibited significant association with the risk of breast to bone metastasis in Pashtun population, while *OPG* (rs2073618) didn't show significant association with disease risk. However, more investigation will be required while targeting the full gene sequencing or whole exome sequencing to find the exact situation of genetic predisposition of breast to bone metastasis in Pashtun Population.

## Supporting information

**S1 Data.**
(XLSX)

**S2 Data.**
(XLSX)

**S3 Data.**
(XLSX)

**S1 Raw images.**
(PDF)

## Author Contributions

**Conceptualization:** Najeeb Ullah Khan, Ahmad M. Alamri, Bushra Iftikhar.

**Data curation:** Faiza Hayat.

**Formal analysis:** Aakif Ullah Khan, Ahmad M. Alamri.

**Investigation:** Faiza Hayat.

**Project administration:** Najeeb Ullah Khan.

**Resources:** Najeeb Ullah Khan, Aakif Ullah Khan, Iftikhar Ahmad, Bushra Iftikhar.

**Supervision:** Najeeb Ullah Khan, Bushra Iftikhar.

**Visualization:** Aakif Ullah Khan, Iftikhar Ahmad.

**Writing – original draft:** Faiza Hayat, Najeeb Ullah Khan.

**Writing – review & editing:** Najeeb Ullah Khan, Aakif Ullah Khan, Ahmad M. Alamri.

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
