## [Decision Letter · Decision Letter 0]

4 Oct 2022

PONE-D-22-23624Risk association of RANKL and OPG gene polymorphism with breast cancer to bone metastasis in Pashtun population of Khyber Pakhtunkhwa, PakistanPLOS ONE

Dear Dr.  Khan

Thank you for submitting your manuscript to PLOS ONE. After careful consideration, we feel that it has merit but does not fully meet PLOS ONE’s publication criteria as it currently stands. Therefore, we invite you to submit a revised version of the manuscript that addresses the points raised during the review process.

Indicate which changes you require for acceptance versus which changes you recommendAddress any conflicts between the reviews so that it's clear which advice the authors should followProvide specific feedback from your evaluation of the manuscriptFor Lab, Study and Registered Report Protocols: These article types are not expected to include results but may include pilot data. 

We look forward to receiving your revised manuscript.

Kind regards,

Filomena de Nigris, Ph.D.

Academic Editor

PLOS ONE

Journal Requirements:

5. Please ensure that you include a title page within your main document. You should list all authors and all affiliations as per our author instructions and clearly indicate the corresponding author.

Reviewers' comments:

Reviewer's Responses to Questions

**Comments to the Author**

1. Is the manuscript technically sound, and do the data support the conclusions?

Reviewer #1: Yes

Reviewer #2: Partly

2. Has the statistical analysis been performed appropriately and rigorously? 

Reviewer #1: Yes

Reviewer #2: Yes

3. Have the authors made all data underlying the findings in their manuscript fully available?

Reviewer #1: Yes

Reviewer #2: Yes

4. Is the manuscript presented in an intelligible fashion and written in standard English?

Reviewer #1: Yes

Reviewer #2: Yes

5. Review Comments to the Author

Reviewer #1: The authors analyze the receptor activator NF-κB ligand (RANKL) and Osteoprotegrin (OPG). The single nucleotide polymorphisms (SNPs) of OPG (rs3102735) and RANKL (rs9533156) exhibited significant association with the risk of breast cancer to bone metastasis in Pashtun populationof Khyber Pakhtunkhwa, Pakistan.

Genetic factors may predispose the probability of breast cancer metastasize to bone.

RANKL-OPG pathway contributes to the primary tumorigenesis and bone metastasis and may be used as a potential prognostic biomarker for breast cancers, in Pashtun population. These conclusions appear to be supported by the results of the present analysis, limited to the population in question.

Reviewer #2: In the manuscript “Risk association of RANKL and OPG gene polymorphism with breast cancer to bone

metastasis in Pashtun population of Khyber Pakhtunkhwa, Pakistan” the authors have associated OPG (rs3102735) and RANKL (rs9533156) polymorphisms with the risk of breast-to-bone metastasis in the Pashtun population.

I recommend accepting this work with Minor revision.

The manuscript is relatively well designed and written despite the group enrolled for the analysis being too small. The statistical analysis was been appropriately performed.

I suggest to the authors some Minor Revisions:

Did you hypotize the relationship between these polymorphisms and the Pashtun population?

Did you evaluate the immune response of patients enrolled and the polymorphisms found? What about the HLA haplotype in these patients? I recommend you this work about bone tumors “ doi:10.3390/

Did you evaluate also the protein expression of RANKL and OPG?

6. PLOS authors have the option to publish the peer review history of their article (what does this mean?). If published, this will include your full peer review and any attached files.

Reviewer #1: No

Reviewer #2: No

---

## [Author Response · Author response to Decision Letter 0]

12 Oct 2022

Dear Editor and Reviewers

Thank you so much for reviewing the manuscript. We believe that the questions you raised will improve our manuscript for broad readership in the field. 

We tried our best to answer your comments. 

Please see the one-by-one responses for the comments below 

Reviewers' comments:

Reviewer #1: The authors analyze the receptor activator NF-κB ligand (RANKL) and Osteoprotegrin (OPG). The single nucleotide polymorphisms (SNPs) of OPG (rs3102735) and RANKL (rs9533156) exhibited significant association with the risk of breast cancer to bone metastasis in Pashtun population of Khyber Pakhtunkhwa, Pakistan.

Genetic factors may predispose the probability of breast cancer metastasize to bone.

RANKL-OPG pathway contributes to the primary tumorigenesis and bone metastasis and may be used as a potential prognostic biomarker for breast cancers, in Pashtun population. These conclusions appear to be supported by the results of the present analysis, limited to the population in question.

Response: Dear reviewer, we really appreciate your time for reviewing the manuscript. Yes, you are right, the basic aim of the study was to see the prognostic potential of the selected polymorphism in breast to bone metastasis in Pashtun population. And this is the first study of this kind in Pashtun population. However, more investigation will be required to confirm the polymorphism in large data sets in the selected population and comparison with other ethnicities. 

Reviewer #2: In the manuscript “Risk association of RANKL and OPG gene polymorphism with breast cancer to bone metastasis in Pashtun population of Khyber Pakhtunkhwa, Pakistan” the authors have associated OPG (rs3102735) and RANKL (rs9533156) polymorphisms with the risk of breast-to-bone metastasis in the Pashtun population.

I recommend accepting this work with Minor revision.

The manuscript is relatively well designed and written despite the group enrolled for the analysis being too small. The statistical analysis was been appropriately performed.

I suggest to the authors some Minor Revisions:

---Did you hypotize the relationship between these polymorphisms and the Pashtun population?

Response: Dear reviewer, first of all we really appreciate your time for reviewing the manuscript. Actually, all the selected participants of the study were Pashtun population of Pakistan. Based on our analysis, OPG (rs3102735) and RANKL (rs9533156) exhibited significant association with breast to bone metastasis while OPG (rs2073618) didn’t show significant association with breast to bone metastasis. The results are well discussed based on previous studies conducted in the discussion part of the study. The OPG (rs2073618) didn’t show association in our data sets but was strongly associated with Egyptians and Chinese Han population cited in the manuscript as [10,14,28,29]. But another study among Caucasians supports our results cited as [30]. However, for exactly hypotize the SNPs with Pashtun population, more recommended more investigation with large data sets in the selected population compared with other ethnicities. 

----Did you evaluate the immune response of patients enrolled and the polymorphisms found? What about the HLA haplotype in these patients? I recommend you this work about bone tumors “ doi:10.3390/

---Did you evaluate also the protein expression of RANKL and OPG?

Response: Dear reviewer, we really appreciate your comments, which will further uncover the downstream process of the SNPs, especially the immune responses against them, changes in genes/protein expression or the downstream molecular and mechanistic approaches after polymorphism. As, this study aimed to see the risk association of the selected polymorphism with breast to bone metastasis, and see weather this polymorphism can be used as prognostic biomarkers for the selected population. Though the points you raised are valid and scientific but it will not come under the main aim of this study. However, future studies would be great to uncover the downstream molecular mechanism including the immune responses, genes/proteins expression, and the protein-protein interaction etc. 

Thank you

---

## [Editor Report · Decision Letter 1]

14 Oct 2022

Risk association of RANKL and OPG gene polymorphism with breast cancer to bone metastasis in Pashtun population of Khyber Pakhtunkhwa, Pakistan

PONE-D-22-23624R1

Dear Dr.Khan

We’re pleased to inform you that your manuscript has been judged scientifically suitable for publication and will be formally accepted for publication once it meets all outstanding technical requirements.

Kind regards,

Filomena de Nigris, Ph.D.

Academic Editor

PLOS ONE